# XPoison: Cross-Class Attacks through Clean-Label Data Poisoning in Fine-Tuning

## Abstract

As deep learning relies on huge datasets for training, poisoning attacks that pollute the datasets pose a significant threat to it security. Given more models pretrained on private corpora inaccessible to external parties, earlier attacks demanding access to the base training datasets have their impact largely diminished, while practical threats focus on the finetuning stage when attackers can accurately target specific (intended) classes by manipulating a small subset of the dataset under their control. Fortunately, attackers could potentially be exposed also thanks to the substantially lowered data volume: e.g., correlation between identities and provided data classes poses risks to attackers. To enable stealthy poisoning, we introduce XPoison that strategically performs poisoning in a *cross-class* manner. Instead of directly poisoning the intended classes, a XPoison attacker only needs to provide dataset for unintended classes and hence hides its identity. We first propose a magnitude matching strategy to more efficiently align the malicious gradients. Furthermore, we estimate contradiction from clean target data and compensate gradient-wise, thereby counteracting its neutralizing influence on the poisoning effect. Through extensive evaluations, we demonstrate that XPoison is capable of robustly reducing the recognition accuracy of targeted classes by up to 38.37% during finetuning, while preserving high accuracy in poison classes.

## 1 Introduction

Deep learning has achieved remarkable successes and been widely deployed across a variety of applications in recent time LeCun et al. (2015). Nevertheless, the reliance of models on large-scale contributed training data makes them inherently vulnerable to attacks, whereby attackers introduce malicious poison to training instances to influence specific model behaviors Nelson et al. (2008). While some earlier studies Geiping et al. (2021) on data poisoning have largely concentrated on models trained from scratch and assumed complete access to the model and its training data, in real-world settings most deployed models are pretrained on private datasets that are inaccessible to external attackers. As models are tasked with increasingly complex problems, finetuning Pan & Yang (2010) provides an effective mechanism for learning from previously unseen domains or refining knowledge in existing domains; however, it also exposes a more practical window for attacks. Early attacks target the specific class associated with the attacker-controlled dataset and involved direct, obvious label modifications, rendering them relatively easy to detect Biggio et al. (2012).

To further enhance stealth and bypass detection mechanisms, attackers instead inject subtle perturbations into the input data while leaving labels unchanged; these modified samples are known as *poisons* Shafahi et al. (2018); Geiping et al. (2021). Nonetheless, this method is not sufficiently stealthy: attacker-controlled finetuning subsets usually map to a few classes, making detection and attribution to the poison and attacker straightforward. Thus, cross-class poisoning is a preferable strategy, as it allows attackers to select misclassification targets at will instead of being limited to the poisoned class, widening the threat surface.

Although it offers greater potential and poses higher risks, developing poison attack algorithms becomes increasingly challenging, as the attack surface requires the model to learn the association between modified inputs and correct labels while keeping the labels fixed. While existing works Shafahi et al. (2018); Huang et al. (2020); Geiping et al. (2021) can achieve decent poisoning effect under specific assumptions, a critical asymmetry between victims and attacks inherently lim-

its such attacks: in real-world model finetuning, attackers typically control only a subset of a data class, while victims may aggregate data from multiple sources and collect clean, correctly-labeled samples of that same class from other providers Russakovsky et al. (2015); Kaissis et al. (2021); Sheller et al. (2020). Such clean data contradicts cross-class poison by providing conflicting or even the exact opposite supervision signal. This raises a natural question: can attackers inject effective biases into models even when correct information of that category is simultaneously present?

Backdoor attacks Gu et al. (2019) can tackle this problem by learning parallel associations where a malicious shortcut to wrong mapping coexists with truthful mapping. This shortcut could be activated by the presence of either a natural Zhao & Lao (2022) or a carefully-optimized Saha et al. (2020) trigger on test-time inputs to affect specific test sample without being canceled out by clean evidence. Because the truthful mapping is still present, normal model performance of correctly classifying inputs without triggers would also not be affected. However, this method loses effectiveness when such trigger isn't available during test-time: attackers may simply not have access to test data for applying triggers or the circumstances determine that they may have little control over the presence of natural triggers in test data Shafahi et al. (2018). This is very likely in practical multi-source data aggregation settings like medical diagnosis or social-media content moderation, where test inputs are directly provided by users Li et al. (2020). The core issue is that building extra pathways based on trigger only temporarily circumvents the contradictory clean information given specific conditions, but it doesn't fundamentally solve the contradiction problem. Whenever the required condition fails, the method no longer suffices. This constraint limits the practicality of clean-label backdoor attacks and makes them inefficient in this specific scenario.

Therefore, trigger-free cross-class poisoning presents a more suitable alternative in this setting, as this can achieve unconditional misclassification without requiring test-time trigger availability while maintaining both efficacy and covertness. Some existing such approaches like Shafahi et al. (2018); Zhu et al. (2019); Aghakhani et al. (2021) work well in transfer learning settings where a pretrained model is finetuned for downstream tasks. Their poison can to certain degree overcome contradiction from clean data either within frozen model knowledge or finetuning dataset. However, they heavily rely on the pretrained feature extractor and cannot work effectively in scenarios where feature space changes significantly due to the large domain difference between finetuning data and pretraining data Shafahi et al. (2018). Other works like Huang et al. (2020); Geiping et al. (2021) have demonstrated promise in controlled settings and doesn't demand feature extractor remaining relatively stable, but they fundamentally assume the non-existence of clean data on target class. This assumption renders them ineffective in realistic multi-source data aggregation setting. A new method is required to overcome this challenge.

In this paper, we introduce XPoison, a novel attack paradigm based on gradient matching that achieves trigger-free cross-class misclassification even when contradictory clean evidence is present. We focus on the scenario of finetuning. In our setting, we assume the attackers know about the model structure.

XPoison employs multiclass gradient matching with two key enhancements: magnitude alignment to match both gradient direction and scale, and contradiction compensation to account for gradient interference from clean data.

Our contributions are:

- Formalizing trigger-free cross-class poisoning as a distinct attack category in concept

- Providing the first evaluation under realistic constraints where finetuning contains clean target information contradictory to poison and existing methods aren't suitable due to failed premises

- Identifying a practical real-world scenario that such attack threatens.

- Introducing clean interference compensation that makes attacks robust to the dilution that occurs when victims train on mixed clean-poison data

This defines a new attack framework for further exploration and reveals a fundamental vulnerability in real-world defense.

The rest of the paper is organized as follows. Section 2 briefly captures related works. Section 3 presents the problem formulation in mathematical terms. Section 4 discusses details of our method. Section 5 reports the experimental results. Finally, section 6 concludes the paper.

## 2 RELATED WORKS

### 2.1 MULTI-SOURCE DATA AGGREGATION

Multi-source data aggregation is the norm of modern ML. From massive webscraping of ImageNet to federated medical systems, many real-world ML systems inherently involve multiple untrusted data sources when collecting training data, which naturally exposes themselves to potential injection of malicious data Russakovsky et al. (2015); Kaissis et al. (2021); Sheller et al. (2020). Security research in federated learning, which takes place in similar setting of distributed data sources converging data to a center, also shows that attacks demonstrate strong effectiveness even with a small percentage of participants being malicious Tolpegin et al. (2020). This validates the practicality of our scenario setting. Such vulnerability to malicious data sources has motivated extensive research into data poisoning attacks, which can be broadly categorized into backdoor attacks and trigger-free attacks based on their activation mechanisms.

### 2.2 BACKDOOR ATTACK

Backdoor attack aims to manipulate model behavior by injecting poisoned data into training and activating in test time through triggers. To increase stealthiness, clean-label backdoor attack is developed to poison data content while maintaining the correct label. For example, Saha et al. (2020) achieves misclassification by crafting poisoned images that appear similar to target-class samples in pixel space while maintaining feature representations close to trigger-patched source images. Zhao & Lao (2022) uses naturally-misclassified samples as poison; Severi et al. (2021) uses SHAP value to select the most important training samples to poison.

There are also works that focus on defending against such attacks or using such attacks for benign purpose. For example, He et al. (2023) trains clean model on poisoned data by isolating the poisoned samples in early training and unlearn them. The key is that stronger attacks are learned faster, which is reflected as bigger and quicker loss drop compared to normal samples Li et al. (2021) uses indiscriminate poisoning attacks to protect unauthorized data usage.

However, all backdoor attacks share a fundamental limitation: they require the presence of trigger during test-time to activate malicious behavior, which makes them infeasible in real-world scenarios like ours.

### 2.3 UNCONDITIONAL ATTACKS

To overcome the limitation of requiring test-time triggers, trigger-free attacks aim to cause unconditional misclassification of unmodified target class through training-time manipulation alone. Shafahi et al. (2018) pioneered this direction by optimizing source-class training images to collide with target-class images in feature space, effectively associating target-class images with source-class labels. Aghakhani et al. (2021) extended this approach by crafting poisons that push target images toward a convex polytope in feature space formed by multiple poison class samples, improving attack transferability across different model architectures. Huang et al. (2020) formulated poisoning as a bilevel optimization problem, using expensive meta-learning to approximate the victim's training process and get optimal poison optimization with unrolled gradient steps. Geiping et al. (2021) simplified this approach through gradient matching, aligning source-class gradients with target misclassification gradients and effectively decreased computational cost.

However, existing unconditional attacks face a critical limitation in realistic deployment scenarios: they assume the victim's training data consists minimal clean data from target class. In practice, victims typically have access to substantial amounts of clean target class data that can dilute or counteract poison effects through correct label associations. This gap between experimental assumptions and real-world conditions motivates our attack that remains effective even when directly-contradictory clean and poisoned data coexist during finetuning.

## 3 PROBLEM FORMULATION

In this section, we present the modeling of the task we aim to address.

### 3.1 THREAT MODEL

We consider a poisoning attack against deep neural networks in the finetuning stage, where attackers seek to manipulate a victim model's behavior through strategic modification of training data.

**Attacker Capabilities:** The attacker can inject a limited number of maliciously crafted training samples into the victim's training set before the fine-tuning process begins. The attacker has no control over the training procedure, model architecture, or labeling process, and must assign correct labels to all injected samples (clean-label constraint).

**Victim Model:** We assume a victim model $f_\theta : \mathcal{X} \to \mathbb{R}^C$ pretrained on a source dataset, which is subsequently fine-tuned on a target dataset $\mathcal{D} = \{(x_i, y_i)\}_{i=1}^N$ where $x_i \in \mathcal{X}$ and $y_i \in \{1, 2, \ldots, C\}$.

**Attack Objective:** Given a specific target sample $(x_t, y_t)$ from the test set, the attacker aims to cause the finetuned model to misclassify $x_t$ as one of any non-$y_t$ classes $\mathcal{Y}_m = \{y_m^{(1)}, y_m^{(2)}, \ldots, y_m^{(k)}\}$ where $y_t \notin \mathcal{Y}_m$, while maintaining the model's performance on clean samples.

### 3.2 MULTICLASS POISONING FRAMEWORK

Unlike traditional single-class approaches, we formulate a multiclass poisoning strategy that leverages samples from multiple poison classes $\mathcal{C}_p = \{c_1, c_2, \ldots, c_k\}$ where $\mathcal{C}_p \cap \{y_t\} = \emptyset$. The attacker selects $n_i$ samples from each poison class $c_i$, creating a poison set:

$$\mathcal{P} = \bigcup_{i=1}^k \mathcal{P}_i, \quad \text{where } \mathcal{P}_i = \{(x_j^{(i)}, c_i)\}_{j=1}^{n_i}. \tag{1}$$

The total poison budget is constrained by $|\mathcal{P}| = \sum_{i=1}^k n_i \leq \epsilon \cdot N$ for some small $\epsilon$ (typically $\epsilon \leq 0.01$). Each poison sample is modified by adding an imperceptible perturbation $\delta_j^{(i)}$ subject to the constraint $\|\delta_j^{(i)}\|_\infty \leq \xi$ for some small $\xi$, yielding poisoned samples $\tilde{x}_j^{(i)} = x_j^{(i)} + \delta_j^{(i)}$.

## 4 METHOD

In this section, we begin by detailing the XPoison framework and subsequently extend it to address the multi-class scenario.

### 4.1 ENHANCED GRADIENT MATCHING

Vanilla gradient matching Geiping et al. (2021) aims to cause unconditional misclassification by matching the gradient of selected poisoned-class samples with that of target-class instances paired with poisoned-class label. Given a target image $x_t$ with true label $y_t$ and intended malicious label $y_m$, the attack optimizes poison samples $\{x_p^{(i)}\}$ with correct labels $\{y_p^{(i)}\}$ to minimize:

$$\mathcal{L}_{\text{direction}} = -\sum_l \langle \nabla_{\theta_l} \mathcal{L}(f_\theta(x_t), y_m), \nabla_{\theta_l} \mathcal{L}(f_\theta(x_p), y_p) \rangle, \tag{2}$$

where $\theta_l$ represents parameters of layer $l$, and the objective maximizes alignment between target and poison gradients.

While directional alignment ensures the poisoned samples point to the same directions as target gradients, vanilla gradient matching suffers from two critical limitations. First, gradients aligning in direction may still differ significantly in magnitude, leading to potentially suboptimal optimization step sizes and consequently poor convergence to attack objectives. Second, during finetuning, clean samples from target class generate contradictory gradients that directly counters the poison signal, which significantly reduces the poison effectiveness as the vanilla approach lacks sufficient strength to overcome the contradiction and achieve effectiveness. As formulated in Eqn. 3, during victim

finetuning, the actual gradient experienced by the model combines poison effects with clean data interference:

$$\mathbf{g}_{\text{total}} = \mathbf{g}_{\text{poison}} + \mathbf{g}_{\text{clean}}. \tag{3}$$

Vanilla gradient matching optimizes for $\mathbf{g}_{\text{poison}} \approx \mathbf{g}_{\text{target}}$, but the actual optimization direction becomes $\mathbf{g}_{\text{poison}} + \mathbf{g}_{\text{clean}}$. This is why poison gradient needs to be enhanced so that clean gradient cannot effectively push toward correct classification.

To address the limitations mentioned, we propose multiple enhancements as solution:

- Magnitude alignment: Explicitly matching the poisoned samples' magnitude with target samples' magnitude in addition to the existing directional alignment. This ensures that poisoned samples not only optimize in the right direction but also update in appropriate strength to overcome potential contradiction from clean samples.
- Clean interference compensation: Estimating the potential impact of clean target data on poison gradient by projection for compensation. This ensures the poison becomes more robust to clean data interference in actual finetuning since clean effects are already approximated in optimization.

Our enhanced objective function becomes:

$$\mathcal{L}_{\text{enhanced}} = \alpha \cdot \mathcal{L}_{\text{direction}}(\mathbf{g}_t^*, \mathbf{g}_p) + \beta \cdot \mathcal{L}_{\text{magnitude}}(\mathbf{g}_t^*, \mathbf{g}_p), \tag{4}$$

where $\mathbf{g}_t^*$ represents the **compensated target gradient** and $\mathbf{g}_p$ represents poison gradients. To compute compensation, we estimate clean interference $\hat{\mathbf{g}}_{\text{clean}}$ using proxy clean samples and subtract that influence from the target gradient:

$$\mathbf{g}_t^* = \mathbf{g}_t - \gamma \cdot \text{proj}_{\hat{\mathbf{g}}_{\text{clean}}}(\mathbf{g}_t), \tag{5}$$

where $\gamma$ controls compensation strength and $\text{proj}_{\hat{\mathbf{g}}_{\text{clean}}}(\mathbf{g}_t)$ represents the projection of target gradients onto the clean interference direction. Due to the significant difference between layers on gradient scales and feature-extraction roles, we apply compensation independently to each layer $l$ for layer-specific projection coefficients:

$$\mathbf{g}_{t,l}^* = \mathbf{g}_{t,l} - \gamma \cdot \frac{\langle \mathbf{g}_{t,l}, \hat{\mathbf{g}}_{\text{clean},l} \rangle}{\|\hat{\mathbf{g}}_{\text{clean},l}\|^2} \hat{\mathbf{g}}_{\text{clean},l}. \tag{6}$$

This removes the component of target gradients that aligns with expected clean interference while preserving magnitude through renormalization, thereby achieving more robust poison against contradictory clean signal. Note that since this is approximation in experimental conditions, samples are randomly selected from the target class in finetuning dataset. Within real-world scenarios, this should be done by sampling representative target class data from external data sources since attackers may not have access to target class data actually used for finetuning. Building on the compensated target gradients, we add explicit magnitude matching in the form of ratio between $\mathbf{g}_p$ and $\mathbf{g}_t^*$

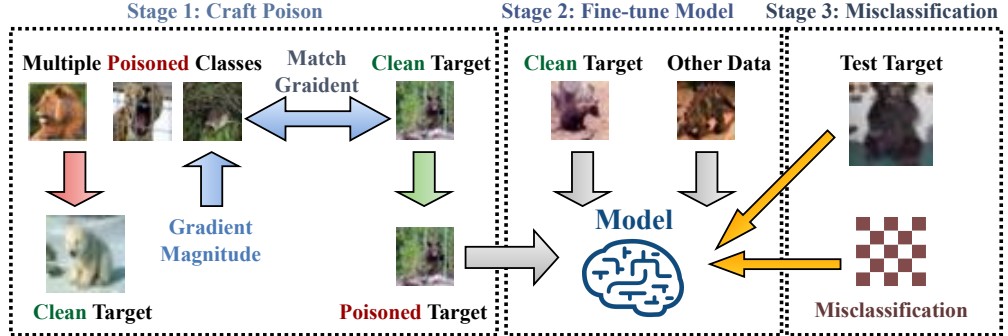

Figure 1: An overview of our poisoning process. Stage 1: Poison samples from multiple classes are crafted to match gradients with a clean target image, enhanced with magnitude alignment and clean interference compensation. Stage 2: The victim model is finetuned on a dataset containing both poisoned samples and clean data. Stage 3: At test time, the target class image is misclassified, demonstrating successful cross-class attack. .

to encourage poisoned gradient to either match or succeed the target gradient for greater poison strength:

$$\mathcal{L}_{\text{magnitude}} = -\log\left(\frac{\|\mathbf{g}_p\|_2}{\|\mathbf{g}_t^*\|_2 + \epsilon}\right). \tag{7}$$

The logarithmic formulation provides several advantages: it naturally handles the vastly different gradient scales across model layers by compressing the value range for consistent processing, prevents optimization instability by bounding extreme magnitude ratios and providing smooth derivatives that avoid sudden jumps during gradient descent, and ensures balanced treatment of both under-magnitude and over-magnitude scenarios for more stable optimization.

## 4.2 MULTI-CLASS POISONING ENHANCEMENT

To further boost the attack effectiveness, we extend our enhanced framework to **multi-class poisoning**, where poisoned samples are distributed across multiple classes instead of one.

Given a set of poison classes $\mathcal{C}_p = \{c_1, c_2, \ldots, c_k\}$, we distribute the poison budget $B$ across all classes:

$$B_i = \left\lceil \frac{B}{|\mathcal{C}_p|} \right\rceil \quad \text{for each class } c_i \in \mathcal{C}_p. \tag{8}$$

We also compute target gradients by averaging across all intended classes rather than using a single intended class:

$$\mathbf{g}_t = \frac{1}{|\mathcal{C}_p|} \sum_{c_i \in \mathcal{C}_p} \nabla_\theta \mathcal{L}(f_\theta(x_t), c_i). \tag{9}$$

This averaged gradient represents the optimization direction toward the centroid of the poisoned class space, providing a potentially more robust and generalizable attack target. Also, the multi-class approach integrates seamlessly with our existing enhancement framework. Clean gradients are estimated independently for each poison class and averaged:

$$\hat{\mathbf{g}}_{\text{clean}} = \frac{1}{|\mathcal{C}_p|} \sum_{c_i \in \mathcal{C}_p} \hat{\mathbf{g}}_{\text{clean}, c_i}. \tag{10}$$

---

**Algorithm 1** Enhanced Gradient Matching with Dual Compensation (Multi-class)

---

**Require:** Target $(x_t, y_t)$, Intended classes $\{y_m^{(j)}\}$, Poison set $\{(x_p^{(i)}, y_p^{(i)})\}$, Model $f_\theta$
**Require:** Parameters: $\alpha, \beta, \gamma$, learning rate $\eta$, iterations $T$
 1: Estimate clean interference: $\hat{\mathbf{g}}_{\text{clean}} \leftarrow \text{EstimateCleanGradients}(x_t, y_t)$
 2: Initialize poison perturbations: $\Delta^{(i)} \leftarrow \mathbf{0}$
 3: **for** $t = 1$ to $T$ **do**
 4:      Apply perturbations: $x_p'^{(i)} \leftarrow x_p^{(i)} + \Delta^{(i)}$
 5:      **Multi-class target gradient:**
 6:         $\mathbf{g}_t \leftarrow \frac{1}{|\{y_m^{(j)}\}|} \sum_j \nabla_\theta \mathcal{L}(f_\theta(x_t), y_m^{(j)})$
 7:      Compensate target gradient: $\mathbf{g}_t^* \leftarrow \text{CompensateGradient}(\mathbf{g}_t, \hat{\mathbf{g}}_{\text{clean}}, \gamma)$
 8:      **for** each poison $i$ **do**
 9:         Compute poison gradient: $\mathbf{g}_p^{(i)} \leftarrow \nabla_\theta \mathcal{L}(f_\theta(x_p'^{(i)}), y_p^{(i)})$
10:         Compute direction loss: $\mathcal{L}_{\text{dir}}^{(i)} \leftarrow -\langle \mathbf{g}_t^*, \mathbf{g}_p^{(i)} \rangle / \|\mathbf{g}_t^*\|$
11:         Compute magnitude loss: $\mathcal{L}_{\text{mag}}^{(i)} \leftarrow -\log(\|\mathbf{g}_p^{(i)}\| / (\|\mathbf{g}_t^*\| + \epsilon))$
12:         Combined loss: $\mathcal{L}_{\text{enhanced}}^{(i)} \leftarrow \alpha \cdot \mathcal{L}_{\text{dir}}^{(i)} + \beta \cdot \mathcal{L}_{\text{mag}}^{(i)}$
13:      **end for**
14:      Update perturbations: $\Delta^{(i)} \leftarrow \Delta^{(i)} - \eta \nabla_\Delta \mathcal{L}_{\text{enhanced}}^{(i)}$
15:      Project perturbations: $\Delta^{(i)} \leftarrow \text{Proj}_\epsilon(\Delta^{(i)})$
16: **end for**
17: **return** $\{x_p^{(i)} + \Delta^{(i)}\}$

---

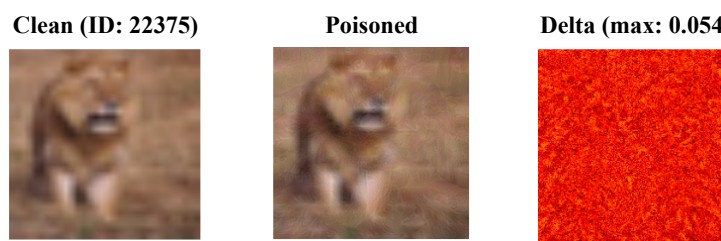

Figure 2: An image example of before and after applying the poison.

It is later applied to the averaged target gradient:

$$\mathbf{g}_t^* = \mathbf{g}_t - \gamma \cdot \text{proj}_{\hat{\mathbf{g}}_{\text{clean}}}(\mathbf{g}_t). \tag{11}$$

The complete algorithm detail can be seen in Algorithm 1. Figure 2 presents an example image of before and after applying the poison.

## 5 EXPERIMENT

In this section, we begin by presenting the experiment setup, followed by a comprehensive evaluation of XPoison.

### 5.1 EXPERIMENT SETUP

For fair comparison with existing baselines, we evaluate on ImageNet-pretrained ResNet18 fine-tuned on CIFAR-100 This represents a realistic transfer learning scenario where attackers target models adapted to new domains or improved on old domains. This setup also highlights the advantages of our gradient-based enhancements even in favorable conditions for feature-based methods.

All our experiments are conducted on two nvidia A5000 GPUs. We set the perturbation budget to 8/225, which means maximum 8 intensity levels are allowed to be changed out of all possible values, making poisoned images visually imperceptible to humans but effectively toxic to model, balancing attack stealthiness with effectiveness. The finetuning learning rate is set to 0.1, selected empirically for optimal performance. Other hyperparameters are kept as default.

We also investigate how pretrained feature representations as clean frozen knowledge affect poisoning transferability by comparing attacks on classes with and without ImageNet overlap. This may better improve the poison effectiveness and isolate the feasibility of poison countering fresh clean target data for finetuning. While pretrained knowledge is absent, poisoned classes include dinosaur, plain, rocket, forest, mountain, sea, shrew, caterpillar, possum, and baby. Target class is cloud. While pretrained knowledge is present, poisoned classes include lion, leopard, tiger, bicycle, motorcycle, shark, whale, spider, fox and elephant. Target class is bear. Note that these experimental configurations involve manual class selection and is an imperfect simulation to the strict domain boundaries typically enforced in real-world deployment.

While design-wise target image is optimized to be misclassified as any of the poisoned classes, in reality, it is acceptable for it to be misclassified as any non-target class.

### 5.2 OVERALL PERFORMANCE

Table 1 shows the comparison between our method and the baseline methods. Compared with clean baseline where no poison is introduced in finetuning, feature-based methods like Poison Frogs Shafahi et al. (2018) and Bullseye Aghakhani et al. (2021), in which poisoned samples are optimized to approach target samples in feature space, have achieved reasonable attack success that reduces target class accuracies by 28.27% and 9.08%, respectively. However, such effectiveness comes with a heavy cost as they are not stealthy enough and cannot maintain decent accuracy on poison class, making them inappropriate for real-world deployment in this scenario. While vanilla gradient matching Geiping et al. (2021) preserves poison class accuracy, it only reduces target class accuracy

by 8.07%, indicating limited attack effectiveness. In contrast, our method achieves a reduction of 38.37%, exceeding the feature-based methods while mostly preserving recognition ability on poison class with a small drop of 2.8%. This indicates the effectiveness and stealthiness of our method.

Table 1: Performance comparison with baselines

| Method | Poison Class Acc (%) | Target Correct (%) |
|---|---|---|
| Clean | 88.00 | 88.88 |
| Poison Frogs | 24.00 | 60.61 |
| Bullseye | 12.00 | 79.80 |
| Gradient matching | 83.30 | 80.81 |
| Ours | 85.20 | **50.51** |

## 5.3 IN-DEPTH ANALYSIS

As shown in Table 2, the best result among all different poison class numbers was achieved in 10 poisoned classes with a drop of 38.37% on target classification rate. While having only one poisoned class, classification rate on poisoned class cannot be maintained well and dropped to 35% while demonstrating suboptimal attack effectiveness. This is likely due to excessive pixel change, which achieved small target misclassification rate at the cost of a complete destruction of poison class recognition. As poison becomes distributed in multiple classes, target recognition linearly drops as poisoned class number increases while preserving a decent level of poison class recognition capability. Due to limited computational resources, we limit the class number within 10.

Table 2: Performance comparison across different numbers of poison classes

| Classes | Poison Class Acc (%) | Target Correct (%) |
|---|---|---|
| Clean | 88.00 | 88.88 |
| 1 | 35.00 | 83.84 |
| 3 | 87.00 | 83.84 |
| 5 | 88.00 | 82.83 |
| 7 | 85.57 | 68.69 |
| 10 | **85.20** | **50.51** |

We also tested when selected classes have no overlap with the pretraining dataset so that the model has no clean frozen knowledge on them. As shown in Table 3, single poison class achieves only 43.00% poison class accuracy despite having a decent 71.72% target correct rate, indicating poor attack stealthiness. After entering multi-class poisoning, poison class accuracy gradually drops from little poison effect in 3 classes to 76.77% with 5 and 7 classes, while in the end slightly increasing back to 80.81% with 10 classes. This U-shaped pattern in target accuracy differs from the linear trend observed in the case with pretrained knowledge, suggesting that the model's adaptation dynamics vary substantially when learning entirely new classes versus refining existing ones. The sustained high poison class accuracy, combined with reasonable attack success rate, indicates that our attack remains effective even without pretrained knowledge present.

Table 3: Comparison across different numbers of poison classes without pretrained knowledge

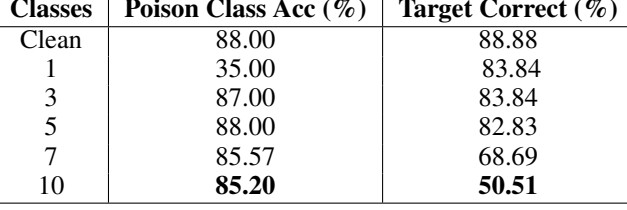

| Classes | Poison Class Acc (%) | Target Correct (%) |
|---|---|---|
| Clean | 88.00 | 88.88 |
| Best | 85.20 | 50.51 |
| 1 | 43.00 | 71.72 |
| 3 | 95.67 | 88.89 |
| 5 | 91.00 | 76.77 |
| 7 | 88.14 | 76.77 |
| 10 | 79.50 | 80.81 |

We also examined the influence of different target class on the attack performance by randomly selecting poison classes from the 10 classes with pretrained knowledge as target. As shown in

Table 4, our method demonstrates consistent performance across different target classes, with poison class accuracy remaining high (75.60-81.60%) and averaging 78.94%. The target correct rates show substantial variation across different target classes, ranging from 55.56% to 73.74% with an average of 67.68%. Target class 88 achieves the strongest attack success with only 55.56% of target samples correctly classified, while maintaining 78.10% poison class accuracy. Target class 43 shows the highest poison class accuracy with 81.60%, but its target correct is at 70.71%, slightly lower than class 88. Overall, this demonstrates that our multiclass gradient matching approach can maintain excellent poison stealthiness and effectiveness regardless of the specific target class chosen.

Table 4: Attack performance across different target classes

| Target Class | Poison Class Acc (%) | Target Correct (%) |
|---|---|---|
| 31 | 79.60 | 73.74 |
| 42 | 75.60 | 65.66 |
| 43 | 81.60 | 70.71 |
| 73 | 79.80 | 72.73 |
| 88 | 78.10 | 55.56 |
| **Average*** | **78.94** | **67.68** |

Table 5 shows our ablation study on enhancement components. We use the best result under 10 poison classes and measure accuracies under different component weight configurations at the same step for fair comparison. $\beta$ represents the weight of magnitude-matching, while $\gamma$ represents the weight of clean compensation. Compared with the baseline of vanilla implementation where both weights are zero, the individual enhancement components demonstrate a modest improvement, with a maximum of 5.05% reduction in target class accuracies. When both are combined, they further result in a substantial 30.30% decrease compared with baseline, confirming the effectiveness of our proposed method.

Table 5: Ablation study

| $\beta$ | $\gamma$ | Poison Class Acc (%) | Target Correct (%) |
|---|---|---|---|
| 1 | 1 | 85.20 | 50.51 |
| 1 | 0 | 77.40 | 77.75 |
| 0 | 1 | 80.60 | 75.76 |
| 0 | 0 | 83.30 | 80.81 |

## 6 CONCLUSION

In this work, we have addressed a critical gap in data poisoning research by examining cross-class attacks under realistic finetuning scenarios where victims possess clean target samples that directly contradict attacker-controlled poisons. The demonstrated effectiveness of trigger-free cross-class poisoning under realistic constraints has significant implications for deployed machine learning systems. Organizations that aggregate finetuning data from multiple sources face previously underestimated vulnerabilities. Even when defenders possess substantial amounts of clean data, sophisticated attackers can still manipulate model behavior through carefully crafted poison samples distributed across a small amount of classes. The formalization of trigger-free cross-class poisoning as a distinct threat category provides a foundation for future defensive research and helps practitioners better assess risks in their ML pipelines.

Our evaluation focuses primarily on image classification tasks using CIFAR-100 in transfer learning scenarios. Future work should extend these findings to other domains, larger-scale datasets, and different model architectures. Additionally, investigating defensive mechanisms specifically designed to handle mixed clean-poison scenarios or cross-class manipulation represents an important research direction.

**Ethics Statement:** This work adheres to the lCLR Code of Ethics and presents data poisoning attacks for research and defensive purposes to advance understanding of ML vulnerabilities and enable better defenses. We do not advocate for malicious use of presented methods.

**Reproducibility:** To ensure reproducibility, we will release the complete source code for verification upon acceptance.

**LLM usage:** This paper used LLM for aiding writing polishing. LLM was not involved in the idealization, method conceptualization, or experimental design. All research concepts and analyses were developed and conducted by the authors.

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
