# OpenReview forum: "XPoison: Cross-Class Attacks through Clean-Label Data Poisoning in Fine-Tuning"
_ICLR.cc/2026/Conference — ICLR 2026 Conference Withdrawn Submission_

### Official Review · Reviewer_CGS6 · 2025-10-31

**Soundness:** 2
**Presentation:** 2
**Contribution:** 2
**Rating:** 2
**Confidence:** 3

**Summary:**

This paper proposes a trigger-free, clean-label, cross-class poisoning attack designed for realistic fine-tuning scenarios where the target class contains a large number of clean samples.Unlike traditional gradient-matching-based poisoning attacks that fail when mixed with clean data, the proposed method introduces two key mechanisms — magnitude alignment and clean interference compensation — to maintain attack effectiveness under mixed clean and poisoned data.
The authors further claim that the proposed cross-class strategy enhances stealthiness by preventing the attacker’s identity from being exposed through the class distribution of contributed data. Experimental results are presented to demonstrate the feasibility of the proposed approach.

**Strengths:**

1.The paper clearly defines the attacker’s capabilities, constraints, and objectives, resulting in a well-scoped and realistic threat model.

2.The main strength lies in proposing a trigger-free poisoning approach that remains effective even when the target class contains a large proportion of clean samples — a setting that closely resembles real fine-tuning pipelines.The introduction of magnitude alignment and clean interference compensation successfully addresses the failure of previous gradient-matching-based poisoning methods in mixed-data environments.

**Weaknesses:**

1.Limited experimental scope:
The experimental setup is relatively simple and lacks broad comparisons with existing poisoning and defense methods. The results, while indicative of feasibility, are insufficient to fully demonstrate the robustness and generalizability of the proposed approach.

2.Insufficient justification of the key motivation:
The paper states in the abstract that “due to a significant reduction in data volume, the attacker’s identity may be exposed through the class distribution of their contributed data”, which serves as the main motivation for proposing a cross-class poisoning attack.
However, this assumption is never elaborated or validated in the main text. Neither Section 1 (Introduction) nor Section 3 (Threat Model) explains how identity exposure arises from class distribution, nor provides empirical, theoretical, or literature-based evidence.
Consequently, the motivation remains largely intuitive and lacks substantiation, making the justification for the “cross-class” design less convincing.

3.Unclear writing and logical flow:
The overall structure and exposition are occasionally unclear. The interaction between key components (i.e., magnitude alignment and clean interference compensation) is not well explained, reducing readability and conceptual clarity. Strengthening logical coherence and presentation would significantly improve the paper’s quality.

**Questions:**

Can you include more comprehensive comparisons with existing clean-label poisoning or gradient-matching methods to better demonstrate the proposed approach’s advantages?

Is there empirical or literature-based evidence supporting the assumption that attacker identity can be inferred from data class distribution?

---

> ### Author Response · Authors · 2025-11-20
>
> Hi. Thank you so much for your comments. This is my first submission attempt in PhD and it does have some problems as you said, thus we decide to withdraw it. Nevertheless, I've learned a lot from your criticism to improve this paper. Wish my next paper would meet higher standards and get accepted. :)

---

### Official Review · Reviewer_xk1y · 2025-10-31

**Soundness:** 2
**Presentation:** 2
**Contribution:** 2
**Rating:** 2
**Confidence:** 2

**Summary:**

This paper proposes XPoison, a trigger-free cross-class clean-label poisoning attack designed for the fine-tuning stage of deep learning models. Instead of poisoning target classes directly, XPoison injects subtle perturbations into other classes to mislead model behavior while hiding the attacker’s identity. The method enhances gradient matching through magnitude alignment and clean interference compensation, making it robust even when clean target data is present. Experiments on CIFAR-100 fine-tuning of ImageNet-pretrained ResNet18 show that XPoison reduces target-class accuracy by up to 38.37% while maintaining high accuracy on poisoned classes, demonstrating strong attack effectiveness and stealth.

**Strengths:**

The paper presents a well-motivated and technically solid contribution by introducing XPoison, a novel trigger-free cross-class clean-label poisoning attack that effectively targets fine-tuned models. Its main strength lies in the innovative enhanced gradient matching mechanism with magnitude alignment and clean interference compensation, which allows the attack to remain effective even when clean target data is present. The proposed method addresses a realistic and underexplored threat scenario in multi-source fine-tuning and demonstrates strong empirical performance, achieving high attack success while maintaining stealth compared to existing baselines like Poison Frogs, Bullseye Polytope, and Gradient Matching.

**Weaknesses:**

1. Threat model inconsistency. The threat model described in the paper assumes that attackers know the model structure and can inject poisons during fine-tuning, while simultaneously claiming no access to the training procedure or architecture. This internal inconsistency makes the assumed attacker knowledge stronger than what is realistic in real-world black-box fine-tuning scenarios, potentially overstating the attack’s practical applicability.
2. The attack is tested under controlled fine-tuning settings, but its persistence under longer training or adaptive defense mechanisms remains unexplored, limiting practical insight into its real-world threat level.
3. The authors note that when fine-tuning data differs substantially from pretraining domains and the feature space changes significantly, existing approaches (including XPoison) degrade in performance. This implies XPoison may fail in common transfer-learning scenarios with large domain shifts, which is a fundamental limitation for real-world applicability.
4. Strong dependence on representative clean samples or high-quality surrogates — The attack requires estimating and compensating for the gradient interference introduced by clean data, which implicitly assumes access to either sufficiently representative target-class clean samples or the ability to train a surrogate model that closely matches the victim. If these conditions are not met (few/no representative samples or surrogate–victim mismatch), the attack success rate is likely to drop substantially; the paper, however, does not evaluate these extreme low-sample or surrogate-mismatch cases to establish failure boundaries.

**Questions:**

1. **Threat Model Consistency**
   The paper assumes attackers know the model structure but later claims the attack works without access to the victim model or training process.
*Could the authors clarify how XPoison operates if the model architecture or gradients are completely unknown?*

2. **Dependence on Clean Samples or Surrogates**
*How does the attack perform when such clean target samples are unavailable or when the surrogate model differs significantly from the victim?*

3. **Robustness to Domain Shift**
 *Can the authors quantify how sensitive XPoison is to domain or feature-extractor changes (e.g., different pretraining datasets or backbones)?*

4. **Scalability and Efficiency**
    *What is the computational cost compared to baseline poisoning attacks? Is the method scalable to large-scale fine-tuning?*

5. **Defensive Robustness**
   No defense or mitigation analysis is provided.
   *Have the authors tested XPoison against common defenses (e.g., data filtering, robust fine-tuning, differential privacy)? How resilient is the attack under such conditions?*

6. **Black-box Realism**
   All experiments appear to assume white-box or high-information settings.
   *Can the authors provide results under more realistic black-box conditions, where the attacker cannot access architecture details or gradients?*

---

> ### Author Response · Authors · 2025-11-20
>
> Hi. Thank you so much for your comments. This is my first submission attempt in PhD and it does have some problems as you said, thus we decide to withdraw it. Nevertheless, I've learned a lot from your criticism to improve this paper. Wish my next paper would meet higher standards and get accepted. :)

---

### Official Review · Reviewer_KvU4 · 2025-10-31

**Soundness:** 1
**Presentation:** 1
**Contribution:** 1
**Rating:** 2
**Confidence:** 4

**Summary:**

This paper studies targeted data poisoning attacks for classification models and considers a "cross-class" scenario in which the attacker selects multiple base classes, rather than a single class, to poison a specific target class. The authors further propose a modified algorithm based on gradient matching attacks and conduct experiments on poisoning during fine-tuning tasks.

**Strengths:**

It is hard for me to find any strength in the paper.

**Weaknesses:**

**Overall**, this paper contains numerous unverified claims that serve as its primary motivations, a poorly justified algorithm that offers limited novelty, and weak experimental validation. I believe this work is substantially below the threshold for ICLR acceptance and requires major revisions before it can be considered for publication.

[W1]: The literature review in this paper is inadequate. While the paper claims to study data poisoning attacks broadly, it only examines four targeted attacks in Section 2.3. The classification of "unconditional attacks" is also poorly defined and should encompass many additional attack types, such as indiscriminate attacks. The authors are encouraged to conduct a more thorough review of the data poisoning literature and include classic attacks in the field. Moreover, a substantial body of work on poisoning attacks against generative models is largely overlooked. Although these attacks may not fit neatly into the proposed framework, they are worth discussing, even if relegated to the appendix. Moreover, Section 2.1 should include more citations and examples on how poison attacks are realistic.

[W2]: It is extremely confusing that the authors devote a substantial portion of the introduction (lines 60-73) and Section 2.2 to discussing backdoor attacks as motivation for targeted attacks. These attack types operate under fundamentally different threat models and employ vastly different algorithms. The practicality of injecting test-time triggers does not provide appropriate motivation for studying training-time attacks, as the two scenarios involve distinct adversarial capabilities and objectives.

[W3]: The authors state in line 84 that existing attacks "fundamentally assume the non-existence of clean data on the target class." This claim is incorrect, as the standard formulation of data poisoning inherently accounts for the presence of clean data. I suggest the authors revisit the bi-level optimization formulation of poisoning attacks to better understand how poisoned and clean data interact during training. Due to this questionable motivation, the analysis presented in Section 4.1 is also unconvincing.

[W4]: Another core motivation of this paper is that using a single base class to generate poisoning samples is inadequate because such attacks are "easily detectable." However, the reasoning behind this claim is unclear. Is detectability due to class imbalance within the training set? If so, a trivial solution exists: inject an equal amount of clean training data into other classes to restore balance. The authors need to clarify what makes single-class poisoning detectable and why their proposed approach offers meaningful advantages over simpler alternatives.

[W5]: The authors introduce "cross-class poisoning" in the title and Section 1 but fail to provide a formal definition in subsequent sections. Section 3.2 then uses the term "multiclass poisoning" instead. Are these terms referring to the same concept?

[W6]: The algorithm proposed seems to break the convergence property of gradient alignment in Geiping et al. Can the authors justify why this method would work better than gradient matching?

[W7]: Fine-tuning with a fixed feature extractor represents a weak threat model that may not reflect realistic attack scenarios. The authors should justify why experiments on training from scratch are not included.

[W8]:  The experimental setup lacks sufficient detail. It appears that the authors are performing class-wise poisoning rather than sample-wise poisoning, but the specific implementation is not clearly described. Moreover, targeted poisoning attacks are typically evaluated using the attack success rate across multiple trials. However, the tables report "poison class accuracy" and "target correctness" instead. The authors should provide clear definitions of these metrics and explain how they relate to standard evaluation protocols in the data poisoning literature.

**Questions:**

Included above.

---

> ### Author Response · Authors · 2025-11-20
>
> Hi. Thank you so much for your comments. This is my first submission attempt in PhD and it does have some problems as you said, thus we decide to withdraw it. Nevertheless, I've learned a lot from your criticism to improve this paper. Wish my next paper would meet higher standards and get accepted. :)

---

### Official Review · Reviewer_69qA · 2025-11-01

**Soundness:** 1
**Presentation:** 1
**Contribution:** 3
**Rating:** 0
**Confidence:** 4

**Summary:**

A data poison attack is proposed. Following the gradient matching method, the proposed method refineds from the following perspectives:
  1. not only directional alignment but also magnitude alignment,
  2. clean interference compensation, which removes the expected clean interference,
  3. considering multi-class poisoning rather than single class.

**Strengths:**

- The proposed method outperforms the baselines in terms of effectiveness and stealthiness. Particularly, it significantly degrades the target acc.
- The ablation study shows that magnitude matching and clean compensation are both effective and stealthy. Furthermore, with these two components, the performance improves by a significant margin.

**Weaknesses:**

- Statistical reporting is missing. Provide multiple runs with variance, confidence intervals, and significance tests to support the results.
- The presentation of the proposed method is not well organized. Since multi-class poisoning is the main contribution of the paper, presenting how the gradients are computed under this consideration first may help readers better understand the mechanism of the proposed method. After introducing the gradient computation, it is recommended to explain why and how clean interference compensation works, followed by a description of the refinement of the objective function (i.e., magnitude matching).
- The proposed method is only evaluated on a single scenario (i.e., ImageNet-pretrained ResNet18 finetuned on CIFAR-100), which is not sufficiently comprehensive.
- The conclusion that target recognition linearly drops as poisoned class number increases is not sufficiently precise.
- The authors state that after entering multi-class poisoning, poison class accuracy gradually drops from little poison effect in 3 classes to 76.77% with 5 and 7 classes, while in the end slightly increasing back to 80.81% with 10 classes. However, this observation should be target acc rather than poison class acc, as shown in Table 3.
- The authors state that target class 43 shows the highest poison class accuracy with 81.60%, but its target correct is at 70.71%, slightly lower than class 88. However, both the acc of class 43 should be higher than those of class 88, as shown in Table 4.

**Questions:**

1. In eq. (8), the resulting sum of $B_i$ may be greater than $B$. How should this case be handled when it occurs?
2. Why does simply averaging the gradient, as in eq. (9), provide a potentially more robust and generalizable attack target?
3. In Figure 2, is the perturbation budget constraint violated, since $0.054>0.036\approx\frac{8}{225}$?
4. The inferior performance of only one poisoned class was attributed to excessive pixel change; however, since the pixel changes were constrained by $\xi$, could the authors elaborate further to clarify this confusion?
5. What does the value "best" in the "classes" column mean in Table 3?

---

> ### Author Response · Authors · 2025-11-20
>
> Hi. Thank you so much for your comments. This is my first submission attempt in PhD and it does have some problems as you said, thus we decide to withdraw it. Nevertheless, I've learned a lot from your criticism to improve this paper. Wish my next paper would meet higher standards and get accepted. :)

---

### Note · Authors · 2025-11-20

I have read and agree with the venue's withdrawal policy on behalf of myself and my co-authors.